Benchmarking Common Factors in Psychotherapy Using AI Systems to Enhance Provider-to-Patient Dynamics to Improve Patient Outcomes

*Alison Cerezo, PhD,[1,2], Vijaykumar Palat, MS[1], Amber Jolley-Paige, PhD[1], Sarah Peregrine Lord, PsyD[1,3]*
*[1]mpathic.ai,  [2]University of California Santa Barbara, [3]University of Washington*

**Background.** Common Factors Theory describes the "key ingredients" of psychotherapy that contribute to positive patient outcomes across various theoretical frameworks (e.g., psychoanalytic, psychodynamic, and cognitive-behavioral therapy, etc.). First developed by researchers like Jerome Frank, and later popularized by Bruce Wampold, Common Factors Theory emphasizes the importance of building a positive therapeutic relationship grounded in empathy, trust, alliance, and the providers' ability to demonstrate competence and unconditional positive regard through active listening - regardless of patient disclosures. While Common Factors Theory is foundational in psychotherapy practice and has proven effective in improving provider-to-patient dynamics across health settings (Wampold, 2015), the application of this theory to the development of AI systems remains relatively unexplored.

In this paper, we present our approach to building and benchmarking an AI system trained in Common Factors Theory using machine learning with natural language processing. Our goal is to explore the integration of this theory into AI-driven psychotherapy communications. Unlike traditional psychotherapeutic practices, the development of AI systems requires a nuanced understanding of how machine learning can encapsulate and apply the principles of Common Factors Theory. This endeavor aims to bridge the gap between psychotherapy theory and the capabilities of machine learning in mental health contexts.

**Method.**
Our team developed mpathic API, an AI-based tool built on 2M+ annotated data, that detects and corrects common factors in clinical skills in real-time (300 milliseconds) to improve 200+ communication behaviors with a specific focus in the life sciences (e.g., health coaching, 911 responders, surgical procedures). We built our initial models using labeled data derived from a coaching/therapy training game trained on responses from coaches and therapists from organizations like the California Rural Indian Health Services, Youper.ai, and the Idaho Crisis Line who learned how to deliver empathic care while earning continuing education credits from the Association for Addiction Professionals. Next, our team's Clinical AI Product team ensured expert clinical annotation of enterprise data in addition to employing synthetic and generative technologies to expand specific labeling strategies and data curation by generating and validating rare use cases. This process began with Common Factors Theory and has expanded to include specialty therapy contexts and adverse events that occur in life science pharmaceutical trials.

For this paper, we describe our approach to benchmarking Common Factors of Empathy and Collaboration on the HOPE dataset—a publicly available dataset comprising 12.8k utterances from 212 therapy sessions involving a therapist and client dyad. Malhotra et al. (2022) conducted thorough processing of the HOPE dataset to eliminate noise and transcription errors. Common Factors Theory encompasses factors from (1) the client, (2) provider and, (3) therapeutic context; we specifically focus on provider behaviors in this paper. Our central research question: Can we produce a scalable, consistent, and unbiased way to assess the occurrences of reflective listening, appreciation, and confrontation–markers of empathy and collaboration, the core features of Common Factors Theory–using natural language processing and AI methods to augment provider communications?

**Preliminary Work.** At mpathic.ai, we developed a universal framework metric for assessing and correcting for common factors in communication behaviors. We created this framework from publicly available research, then developed and validated for acceptability with several industry advisors and

aligned it to the literature. Human raters and annotators reviewed publicly available as well as private session transcripts from enterprise customers at the sentence level for specific behaviors following the framework. Raters were trained to reliably detect behaviors within utterance-level agreement above .8 (Krippendorff's alpha) (Note: This is some of the highest interrater reliability reported in the literature, as most studies report ICCs above .8 using conversation-level agreement). Machine learning methods were used to create natural language processing models based on conversational training data (i.e., transcripts of patients and provider dyads; employee and supervisor dyads). This data was augmented with synthetic training examples (i.e., conversations from gold-standard facilitator role plays and expert-generated data via LLMs).

**Findings to be Reported:** The models will be evaluated on novel test and inference data with F scores comparable to or better than human raters, showing that AI can successfully detect and coach providers to foster common factors communications skills with the end goal of improving relational alliance and patient outcomes. In pilot work, we found that our API demonstrated 93% accuracy in detecting Appreciation and 100% accuracy in detecting Toxicity and Confrontation when compared to gold-standard human annotators (i.e., mental health clinicians, including licensed psychologists) on a health customer dataset. For this paper, we will extend our pilot work by benchmarking the Common Factors of Empathy and Collaboration on the HOPE dataset, annotated to capture therapist behavior and extending our analysis to include: (1) API performance against gold standard human annotators; (2) precision/recall via F1 scores.

**Conclusion.** Common Factors encompass critical relational dynamics that foster trust and collaboration between the provider-patient dynamic that result in positive patient outcomes. While Common Factors have been demonstrated to be a key ingredient in the delivery of effective mental health care, the application of Common Factors in AI development remains relatively unexplored. An important next step in the development of AI in the health space is benchmarking the accuracy of AI models to detect, and consequently coach, providers in Common Factors—a key ingredient in patient care.

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
