# OpenReview forum: "Common Factors in Psychotherapy: Enhancing Provider-to-Patient Dynamics to Improve Patient Outcomes"
_AAAI.org/2024/Spring_Symposium_Series/Clinical_FMs — AAAI 2024 SSS on Clinical FMs_

### Official Review · Reviewer_AfmP · 2024-02-23
**Review for Common Factors in Psychotherapy: Enhancing Provider-to-Patient Dynamics to Improve Patient Outcomes**

**Rating:** 4
**Confidence:** 4

**Review:**

Summary of the paper:
The abstract describes the use of NLP to detect healthcare provider behaviors aligned with common factors theory in psychotherapy. Common factors theory emphasizes building empathy, trust and positive relationships through provider skills like reflective listening and appreciation. While the clinical importance of this is paramount, I have concerns with the content that has been presented in the abstract.

Major Comments:

This is a great problem statement. However, I have a couple of major comments:

1. There is ambiguity in the description of the exact methods, vague terms such as machine learning and natural language processing are used.

* They mention the use of “synthetic and generative technologies to expand specific labeling strategies and data curation by generating and validating rare use cases” but don’t describe this - what was the generative model that was used, how did they verify its realism to simulate rare use cases, how much synthetic data was generated relative to non-synthetic data etc. More details need to be provided since this can have significant impacts on the quality of their model.

* “Machine learning methods were used to create natural language processing models based on conversational training data”. What was the base NLP model, did the authors fine-tune a model such as LLaMA? What specific machine learning method was used - NLP fine-tuning strategy needs to be described.
2. The authors mention that they will report results of benchmarking their model on the HOPE dataset, but they don’t do so within the paper.

The overall clarity of the abstract is low due to the above concerns.

Minor Comments:

1. They have not adhered to the AAAI submission format.
2. They use the phrase “using machine learning with natural language processing”. NLP is technically a subfield of ML, and this statement needs to be revised to reflect that.

---

### Official Review · Reviewer_9KXZ · 2024-02-24

**Rating:** 3
**Confidence:** 4

**Review:**

This abstract presents an approach to building and benchmarking an AI system trained on Common Factors Theory using NLP techniques. Using labeled data, the work aims to use synthetic and generative technologies to generate and validate rare user cases. However, the model/method details are not clearly stated.

Pros:
* Applying common factors in AI development is a relatively novel idea
* Data in use is comprehensive and potentially sufficient

Cons:
* Method detail is not clearly stated (model architecture, how training is done etc.), and how Common Factors is integrated/reflected is not clear from current writing
* What is the difference between this work and the work supervised to detect different common factors? It would need more work to distinguish the contribution of this work and existing works.
* Vague connection with foundation models

---

### Official Review · Reviewer_KZ6p · 2024-02-24
**Interesting work but major corrections are required**

**Rating:** 5
**Confidence:** 4

**Review:**

The work "Benchmarking Common Factors in Psychotheraphy Using AI Systems to Enhance Provider-to-Patient Dynamics to Improve Patient Outcomes" is a really interesting paper that comprises a kind of a foundation model for neuro-symbolic AI connected with physchotherapy. However, there are several points that need to be clearly addressed before the work can be accepted for publication. To be clear enough, the general reviewer remarks are given in the form of numbered list provided below.

1. The Authors in the section "Method" wrote "(...) We built our initial models using labeled data derived from (...)". This statement is not really clear. The Authors need to address what was the structure of the data and how much of them were taken into account when it comes to the training/testing dataset. Right now, the overview of the data is missing.
2. In section "Method" one can also read that the Authors used methodologies for generation of the synthetic samples. However, once again, the description of the used algorithms and methods is missed. It is unclear what was the approach to generate these samples. The Authors must specify what kind of AI models or methodologies were consumed to generate the synthetic data. On the other hand it is also unclear how much samples were generated in that manner. This information is also needed to appropriately validate the worked-out models.
3. Section "Preliminary work" - there is information that Machine Learning models were used to create NLP algorithms. However, once again the details are missing. The Authors have to provide information about the algorithms that were used in their approach. Right now, it is impossible to understand the approach.
4. "Findings to be Reported" - one can read information about the levels of accuracy in detection of Appreciation and Toxicity/Confrontation - however, once again there is no sufficient details. How the algorithms were evaluated? What kind of approach was used for this aim? How the database was split and how much samples were consumed. All these questions need to be addressed.

To sum up, I would like to recommend the work for publication but only after major revision that will address all the statements given above.

---

### Official Review · Reviewer_QiAY · 2024-02-27
**Benchmarking Common Factors in Psychotherapy Using AI Systems to Enhance Provider-to-Patient Dynamics to Improve Patient Outcomes**

**Rating:** 6
**Confidence:** 3

**Review:**

Authors present the use of LLM to improve psychotherapy sessions integrating common factors approach into LLM to provide feedback.
The paper is certainly original. It would enhance the clarity and understanding of the paper if authors present more detail on how the system is built, the interpretation of the metrics and how the system can improve the quality of visits. I would also appreciate a discussion on the ethical or social implications of using this type of technology in the medical setting.